# Dysregulation of Dopaminergic Regulatory Factors TH, Nurr1, and Pitx3 in the Ventral Tegmental Area Associated with Neuronal Injury Induced by Chronic Morphine Dependence

**DOI:** 10.3390/ijms20020250

**Published:** 2019-01-10

**Authors:** Weibo Shi, Yaxing Zhang, Guoting Zhao, Songjun Wang, Guozhong Zhang, Chunling Ma, Bin Cong, Yingmin Li

**Affiliations:** Hebei Key Laboratory of Forensic Medicine, Collaborative Innovation Center of Forensic Medical Molecular Identification, Department of Forensic Medicine, Hebei Medical University, Shijiazhuang 050017, China; shiweibo56@hotmail.com (W.S.); 15369305705@163.com (Y.Z.); zhaoguoting33@126.com (G.Z.); wangsongjun@hebmu.edu.cn (S.W.); zhanggz3@126.com (G.Z.); chunlingma@126.com (C.M.)

**Keywords:** chronic morphine dependence, ventral tegmental area, dopaminergic neurons, TH, Nurr1, Pitx3

## Abstract

The ventral tegmental area (VTA), a critical portion of the mesencephalic dopamine system, is thought to be involved in the development and maintenance of addiction. It has been proposed that the dopaminergic regulatory factors TH, Nurr1, and Pitx3 are crucial for determining the survival and maintenance of dopaminergic neurons. Thus, the present study investigated whether abnormalities in these dopaminergic regulatory factors in the VTA were associated with neuronal injury induced by chronic morphine dependence. Rat models with different durations of morphine dependence were established. Thionine staining was used to observe morphological changes in the VTA neurons. Immunohistochemistry and western blot were used to observe changes in the expression of the dopaminergic regulatory proteins TH, Nurr1, and Pitx3. Thionine staining revealed that prolonged morphine dependence resulted in dopaminergic neurons with edema, a lack of Nissl bodies, and pyknosis. Immunohistochemistry showed that the number of TH^+^, Nurr1^+^, and Pitx3^+^ cells, and the number of TH^+^ cells expressing Nurr1 or Pitx3, significantly decreased in the VTA after a long period of morphine dependence. Western blot results were consistent with the immunohistochemistry findings. Chronic morphine exposure resulted in abnormalities in dopaminergic regulatory factors and pathological changes in dopaminergic neurons in the VTA. These results suggest that dysregulation of dopaminergic regulatory factors in the VTA are associated with neuronal injury induced by chronic morphine dependence.

## 1. Introduction

Morphine is a common analgesic drug that is highly efficient and is highly addictive. Abuse of morphine has greatly increased in recent years [1,2], leading to many negative effects on individuals and society and to significant medical and public health problem. The ventral tegmental area (VTA), a critical part of the mesencephalic dopamine system, participates in almost all of the rewarding effects of drug dependence and is thought to be the key brain region involved in the development and maintenance of addiction [3,4]. As broadly documented in the literature, morphine administration contributes to multiple adaptive changes in molecular and cellular function in the mesencephalic dopamine system [5,6], the remodeling of neural structures [7], and drug resistance in addicts [8], which are thought to be linked to persistent cravings and relapse. 

Among the transcription factors involved in the development and physiological function of mesencephalic dopaminergic neurons, tyrosine hydroxylase (TH), nuclear receptor related factor 1 (Nurr1), and pituitary homeobox 3 (Pitx3) play a critical role in many aspects of neuronal physiology [9,10,11]. TH, the rate-limiting enzyme in dopamine synthesis [12], is involved in neurotransmission, determining dopamine transmitter identity, and the survival of dopaminergic neurons [9,12]. Nurr1 is an orphan member of the nuclear receptor superfamily of transcription factors and is critical for the differentiation, migration, maturity, and survival of dopaminergic neurons in the mesencephalon [13,14], as well as being partially involved with non-dopaminergic neurons [15,16,17]. It is also essential for the transcription of sets of genes, including *Th*, involved in dopaminergic neuron metabolism [9,12]. Pitx3 is another critical transcription factor for dopaminergic neurons. The gene encoding Pitx3 is expressed exclusively in mesencephalic dopaminergic neurons and activates the transcription of genes directly involved in the survival and maintenance of these neurons. However, it remains unclear what effects TH, Nurr1, and Pitx3 have on dopaminergic neurons relating to morphine dependence. To date, only two related studies have been published. A study by Horvath et al. [18] showed the relationship between Nurr1 and opioid receptors in individuals addicted to heroin. Further, García-Pérez et al. [19] described changes in TH, Nurr1, and Pitx3 in specific brain regions after one week of morphine exposure and morphine withdrawal in rats, but the changes after longer morphine dependence remain unclear. Therefore, in order to simulate human characteristics after long-term morphine exposure, we administered morphine to morphine-dependent rats for up to six weeks and investigated whether abnormalities in TH, Nurr1, and Pitx3 in the VTA are associated with dopaminergic neuron injury induced by chronic morphine dependence. The findings of the present study provide evidence of the mechanisms of nerve injury induced by chronic morphine dependence.

## 2. Results

### 2.1. Thionine Staining Showed Pathological Changes in VTA Dopaminergic Neurons

In the control group, the neuronal structure was clearly visible and the Nissl bodies were evenly distributed in the cytoplasm. After 1 week of morphine dependence, edema was visible in the neurons, and Nissl bodies were unevenly distributed. After 3 weeks of morphine dependence, some Nissl bodies disappeared and pyknotic neurons were visible. Cellular damage was more obvious, Nissl bodies disappeared, and neurons were pyknotic and dying at six weeks of morphine dependence (Figure 1).

### 2.2. TH Expression in the VTA

The cytoplasm and synapses of dopaminergic neurons in the VTA were conspicuously marked by the anti-TH antibody (Figure 2A). ANOVA for the TH^+^ cells in the VTA revealed that there were significant differences among the groups (*F*_[3,28]_ = 113.161; *P* < 0.001). Compared with the control group (193.88 ± 5.71), the number of TH^+^ cells was significantly lower after three weeks (134.63 ± 4.18, *P* < 0.05) and six weeks (93.75 ± 2.76, *P* < 0.05) of morphine dependence, although there was no difference at one week of dependence (183.13 ± 4.24, *P* > 0.05) (Figure 2B). 

### 2.3. Western Blot Analysis of TH Expression

Consistent with the immunohistochemistry results, the relative level of TH in the VTA decreased with prolonged morphine dependence. Compared with the control group, there was no significant difference in the one week morphine-dependent group (*P* > 0.05), but TH expression in the three and six week morphine-dependent groups markedly decreased (*P* < 0.05 and *P* < 0.01, respectively) (Figure 3).

### 2.4. Nurr1 Expression in the VTA

Double labeling showed that Nurr1 was highly co-localized with the dopaminergic neuron marker TH in the VTA, as shown in Figure 4A. ANOVA for TH^+^, Nurr1^+^, and Nurr1^+^-TH^+^ cells in the VTA showed that there were significant differences among the groups (TH^+^, *F*_[3,28]_ = 100.920; *P* < 0.001; Nurr1^+^, *F*_[3,28]_ = 133.173; *P* < 0.001; Nurr1^+^-TH^+^, *F*_[3,28]_ = 59.348; *P* < 0.001). As depicted in Figure 4B–D, the number of TH^+^, Nurr1^+^, and Nurr1^+^-TH^+^ cells significantly decreased after three weeks (TH^+^, 135.25 ± 4.24, *P* < 0.05; Nurr1^+^, 147.50 ± 4.51, *P* < 0.05; Nurr1^+^-TH^+^, 126.25 ± 5.91, *P* < 0.05) and six weeks (TH^+^, 93.75 ± 3.01, *P* < 0.05; Nurr1^+^, 101.13 ± 2.45, *P* < 0.05; Nurr1^+^-TH^+^, 88.00 ± 3.64, *P* < 0.05) of morphine dependence compared with the control group (TH^+^, 192.38 ± 6.08; Nurr1^+^, 203.38 ± 4.90; Nurr1^+^-TH^+^, 181.13 ± 6.65), although there was no difference after one week of dependence (TH^+^, 182.13 ± 4.24, *P* > 0.05; Nurr1^+^, 197.63 ± 4.33, *P* > 0.05; Nurr1^+^-TH^+^, 175.63 ± 6.19, *P* > 0.05).

### 2.5. Western Blot Analysis of Nurr1 Expression

The relative level of Nurr1 in the VTA after one week of morphine dependence increased compared with the control group (*P* < 0.05). However, with prolonged morphine exposure, the relative Nurr1 expression in the morphine-dependent groups at three and six weeks decreased significantly (*P* < 0.05) (Figure 5).

### 2.6. Pitx3 Expression in the VTA

The Pitx3 expression pattern was similar to that of Nurr1 in TH^+^ VTA neurons. Immunofluorescence double staining showed that Pitx3 was highly co-localized with TH^+^ cells in the VTA (Figure 6A). ANOVA for TH^+^, Pitx3^+^, and Pitx3^+^-TH^+^ cells in the VTA showed that there were significant differences among the groups (TH^+^, *F*_[3,28]_ = 139.213; *P* < 0.001; Pitx3^+^, *F*_[3,28]_ = 140.801; *P* < 0.001; Pitx3^+^-TH^+^, *F*_[3,28]_ = 70.301; *P* < 0.001). As shown in Figure 6B–D, compared with the control group (TH^+^, 198.50 ± 4.81; Pitx3^+^, 208.00 ± 5.37; Pitx3^+^-TH^+^, 184.63 ± 6.05), the numbers of TH^+^, Pitx3^+^, and Pitx3^+^-TH^+^ cells were significantly lower in the morphine-dependent groups at three weeks (TH^+^, 133.88 ± 4.32, *P* < 0.05; Pitx3^+^, 149.13 ± 3.71, *P* < 0.05; Pitx3^+^-TH^+^, 126.25 ± 5.01, *P* < 0.05) and six weeks (TH^+^, 90.25 ± 3.05, *P* < 0.05; Pitx3^+^, 99.88 ± 3.46, *P* < 0.05; Pitx3^+^-TH^+^, 84.88 ± 4.27, *P* < 0.05), while no difference was detected at one week (TH^+^, 187.50 ± 4.63, *P* > 0.05; Pitx3^+^, 198.50 ± 4.11, *P* > 0.05; Pitx3^+^-TH^+^, 174.88 ± 6.45, *P* > 0.05).

### 2.7. Western Blot Analysis of Pitx3 Expression

The expression of Pitx3 was significantly upregulated after one week of morphine dependence (*P* < 0.05). However, after prolonged morphine dependence (three and six weeks), the level of Pitx3 significantly decreased (*P* < 0.05 for both groups) (Figure 7).

## 3. Discussion

The mesencephalic dopamine system is the key brain region involved in the development and maintenance of addiction [3,4]. Mesencephalic dopaminergic neurons are primarily distributed in the VTA. Thus, changes in VTA neurons likely reflect the effects of drugs of dependence, including morphine, on the dopaminergic system. Substantial studies have indicated that morphine administration results in dysfunction of mesencephalic dopaminergic neurons [20,21] and remodeling of the nerve structure [22]. However, the changes of dopaminergic neurons in the mesencephalon during chronic morphine dependence have not been clearly shown. In relation to our previous study [23], the present study showed that damage to dopaminergic neurons could be clearly observed during prolonged morphine exposure.

Multiple adaptive changes in molecular and cellular function in the mesencephalic dopamine systems have been shown after acute and chronic morphine administration, which are thought to be associated with the persistent cravings and relapses in addicts [24,25]. TH, Nurr1, and Pitx3 play critical roles in determining dopamine transmitter identity and neurotransmission, as well as in the survival and maintenance of dopaminergic neurons [9,10,11]. TH, the rate-limiting enzyme in dopamine synthesis, has been widely recognized as a specific marker of dopaminergic neurons [9,12]. Therefore, changes in TH expression can directly reflect the changes in dopaminergic neurons. The binding sites for Nurr1 and Pitx3 have been identified at the promoter region of the *Th* gene [12,26,27]; they thereby regulate TH and promote the maintenance of dopaminergic neurons during adult stages [6,28]. Very few TH^+^ cells that are not Nurr1^+^ [11] or Pitx3^+^ [29] can be observed in the VTA, consistent with our present findings. The data suggest that Nurr1 and Pitx3 are critical regulatory factors for dopaminergic neurons.

Morphine targets gamma-aminobutyric acid (GABA)-ergic neurons, via the binding of μ-opioid receptors (MOR), and decreases their activity, which leads to an indirect upregulation of dopaminergic neuron activity and dopamine release into synaptic clefts in the VTA [30]. García-Pérez et al. [19] indicated that the expression of Nurr1 and Pitx3 significantly increased while TH remained unchanged during one week of morphine dependence, which is consistent with the present findings. These data suggest that a regulatory interaction exists between these transcription factors and morphine in the mesencephalic dopamine system. Morphine increases the levels of these transcription factors through the regulation of a series of complex neural pathways, which ultimately maintains the homeostasis of dopaminergic neurons. The longest period of morphine exposure in previous studies was one week and the changes during longer morphine dependence needed to be explored. For the present study, we administered morphine to morphine-dependent rats for up to six weeks, for the purpose of simulating human characteristics under long-term morphine exposure, and investigated the relationship between changes in these transcription factors and injury to the VTA dopaminergic neurons. The results of the present study indicated that the expression of TH, Nurr1, and Pitx3 significantly decreased and damage to dopaminergic neurons was clearly visible after prolonged morphine dependence. Although the contribution of regulatory mechanisms to chronic morphine dependence is not fully understood, we hypothesized, based on the known mechanisms of opioid addiction, that the prolonged consumption of morphine would result in a decrease in sensitivity of GABAergic neurons and changes in the activity of dopaminergic neurons. In addition, we proposed that anxiety during long-term drug cravings leads to hypo-functionality of opioid receptors as well as corresponding changes in dopaminergic transmitters in terminal fields [31,32], which all contribute to changes in dopaminergic regulation factors that determine the survival of dopaminergic neurons. Changes in these transcription factors may directly or indirectly effect synaptic transmission, plasticity, and synaptic maintenance in dopaminergic neurons throughout adulthood [33,34,35,36]. Therefore, we think that a significant decrease in dopaminergic regulation factors may be a primary cause of dopaminergic neuron injury induced by chronic morphine dependence.

In conclusion, the present study clearly indicates that a significant reduction in expression of the dopaminergic regulation factors Nurr1 and Pitx3 could not effectively regulate TH in the VTA during chronic morphine dependence, which affects the survival of dopaminergic neurons and ultimately leads to neuronal pathology. These novel findings provide morphological evidence that dysregulation of dopaminergic regulatory factors in the VTA are associated with neuronal injury induced by morphine dependence.

## 4. Materials and Methods

### 4.1. Reagents

Rabbit monoclonal anti-TH antibody, mouse monoclonal anti-Nurr1 antibody, mouse monoclonal anti-Pitx3 antibody and anti-β-actin antibody were purchased from Abcam (Cambridge, UK). The immunohistochemical kit was purchased from Beijing Zhongshan Goldenbridge Biotech (Beijing, China), DyLight^TM^ 488-Conjugated AffiniPure goat anti-mouse IG and DyLight^TM^ 594-Conjugated AffiniPure goat anti-rabbit IG were purchased from Jackson ImmunoResearch (West Grove, PA, USA), and the morphine hydrochloride injection (10 mg/mL) was produced in the first pharmaceutical factory of Shenyang (Shenyang, China).

### 4.2. Animals

Adult male Wistar rats (Experimental Animal Center, Hebei Medical University, China), weighing 250 ± 20 g were housed (3/cage) in a 12:12 h light/dark cycle room. The rats were given free access to food and water. All the experiments were approved on 26/10/2015 by the University of Hebei Medical Branch Institutional Animal Care and Use Committee (IACUC-Hebmu-2015263). The rats were randomly assigned into the following groups: control (Con) group, and the 1 week, 3 week, and 6 week morphine (Mor) dependence groups (*n* = 12 rats per group).

### 4.3. Models of Morphine Dependence

The models of morphine dependence were established as previously described [23]. In brief, morphine hydrochloride was injected subcutaneously into the back of morphine-dependent rats twice daily (8:00, 20:00) for 5 days. The initial dose was 10 mg/kg and it was increased by 10 mg/kg every other day. The control group received an equal volume of saline. After 5 days of administration, the rats were confirmed to be dependent on morphine. This process, described by Maldonade [37], involved observation for signs of opiate withdrawal including stretching, standing, jumping, wet-dog shakes, cleaning fur, teeth chattering, and swallowing (Table 1). After morphine dependence, the rats were given a morphine injection of 30 mg/kg twice daily until the 1, 3, or 6 week mark.

### 4.4. Tissue Preparation

Two hours after their last morphine or saline injection, the rats were deeply anesthetized and sacrificed. The brains used for staining were harvested and fixed immediately in 10% formalin. Tissues were subsequently dehydrated in a graded ethanol series and embedded in paraffin. Brain sections (5 mm) beginning at −4.80 mm from the bregma were obtained with the aid of a stereotaxic atlas (Figure 8a) [38]. They were then prepared for thionine staining and immunohistochemical staining and examined under light microscope (Olympus IX71; Olympus, Tokyo, Japan). The brain samples for western blot were removed and immediately frozen in liquid nitrogen. According to the rat brain atlas of Paxinos and Watson [38], the samples of the VTA were accurately extracted using a hole-punch device with a 1 mm internal diameter (Figure 8b).

### 4.5. Immunohistochemistry

The experiments were performed according to the manufacturer’s instructions. Deparaffinized sections were pretreated by performing microwave antigen retrieval, submerging them in 3% H_2_O_2_ cold methanol for 30 min, and then in goat serum working liquid for 30 min. For detection of TH^+^ cells, the tissues were incubated overnight with antibodies against TH (1:200) at 4 °C. After being washed with PBS, the sections were incubated for 1 h with the biotinylated secondary antibody and subsequently incubated with alkaline phosphatase (AP)-conjugated biotin for 30 min. Finally, AP-red was applied to the sections for 5 min. For visualizing the locations of immunostaining, the tissues were counterstained with hematoxylin.

### 4.6. Immunofluorescence Double Staining

Immunofluorescence was performed as described previously [17]. Antibodies against Nurr1 (1:100) or Pitx3 (1:100) were used as the first primary antibodies. The anti-TH antibody (1:200) was used as the second primary antibody. DyLight^TM^ 488-Conjugated AffiniPure goat anti-mouse IG (1:100) and DyLight^TM^ 594-Conjugated AffiniPure goat anti-rabbit IG (1:150) were used as secondary antibodies. 

### 4.7. Cell Counting

Eight rats from each group were used for morphological observation and data analysis. According to the stereotaxic atlas [38], the largest VTA was accurately exposed (Figure 8c). Using the serial sectioning technique, we took every fifth section, for a total of three sections for each rat. The numbers of TH^+^, Nurr1^+^, Pitx3^+^, Nurr1^+^-TH^+^, and Pitx3^+^-TH^+^ cells were counted at 100 times magnification. The average number of positive cells in each rat was calculated by two independent observers who were blind to the experimental conditions.

### 4.8. Western Blot Analysis

Four rats from each group were used for western blot. According to a previously described protocol [20], tissue extracts (50 μg of protein/lane) were loaded into an SDS-PAGE gel, separated by electrophoresis, and transferred to polyvinyl difluoride (PVDF) membranes. The membranes were incubated overnight at 4 °C with anti-TH (1:500), anti-Nurr1 (1:200), anti-Pitx3 (1:200), and anti-β-actin antibodies. The membranes were incubated with goat anti-mouse/rabbit IgG horseradish peroxidase-conjugate and then exposed to X-ray film by an enhanced chemiluminescence system. LabWorks 4.5 software (LabWorks, UT, USA) was used to measure the intensity of the bands.

### 4.9. Statistical Methods

The Kolmogorov–Smirnov test showed that the data were normally distributed in all groups (*P* > 0.1). The results are presented as mean ± SEM. Statistical analysis was performed by one-way ANOVA. The significance was defined as *P* < 0.05 for all statistical tests.

## Figures and Tables

**Figure 1 ijms-20-00250-f001:**
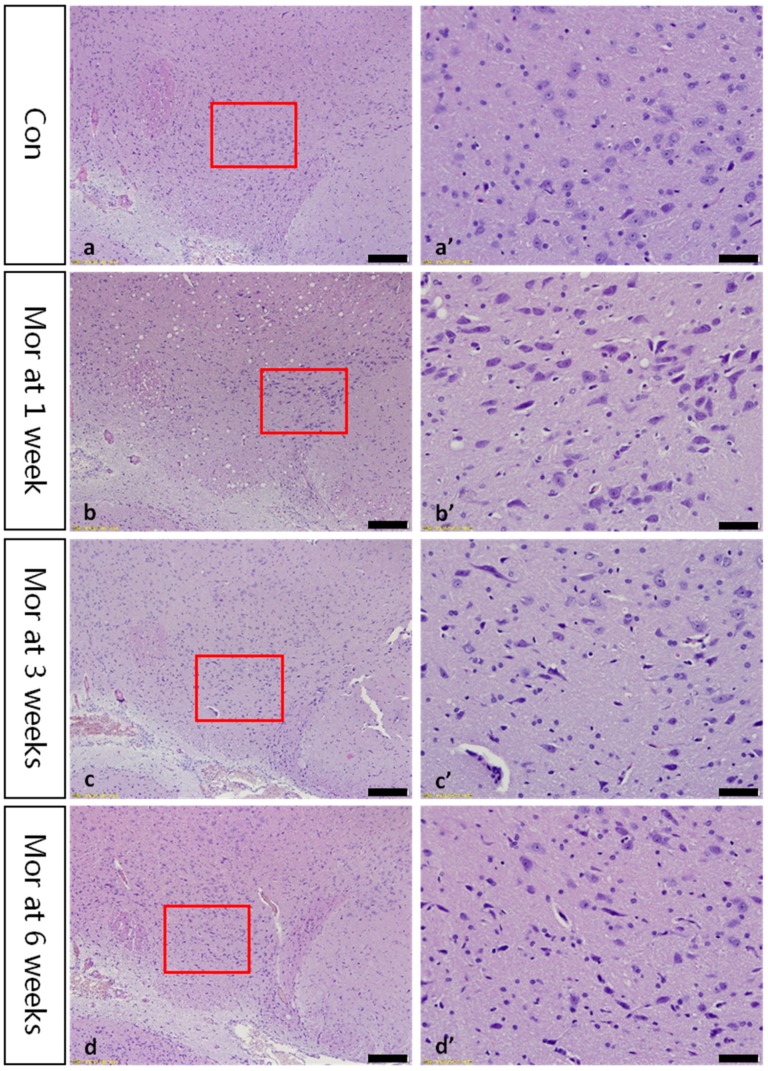
Thionine staining of the VTA. (**a’**–**d’**) are magnified areas of (**a**–**d**), respectively. With prolonged of morphine dependence, Nissl body structures are not clear and neurons are pyknotic and deeply stained. Bars = 100 μm in (**a**–**d**); Bars = 50 μm in (**a’**–**d’**).

**Figure 2 ijms-20-00250-f002:**
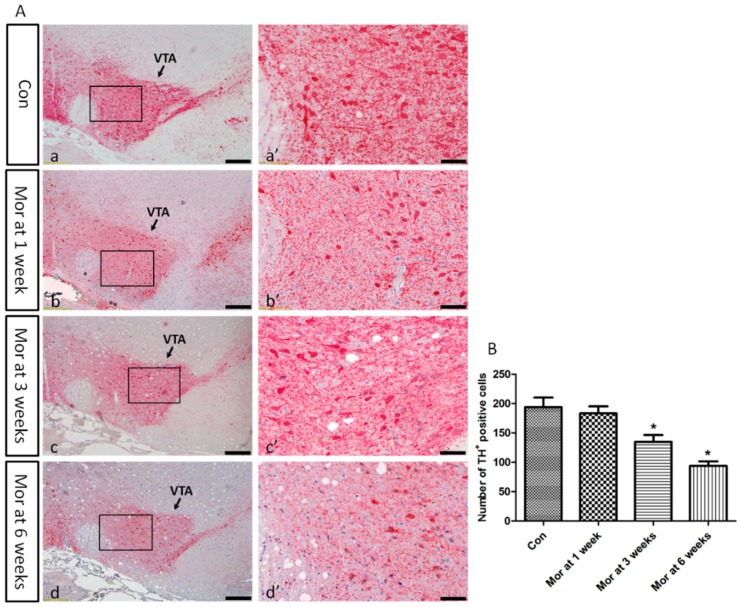
(**A**) Representative images showing TH immunohistochemistry in the VTA. (**a’**–**d’**) are magnified areas of (**a**–**d**), respectively. Bars = 100 μm in (**a**–**d**); Bars = 50 μm in (**a’**–**d’**). (**B**) Quantitative analysis of TH^+^ positive cells. The data are shown as mean ± SEM, * *P* < 0.05 vs. control group (*n* = 8).

**Figure 3 ijms-20-00250-f003:**
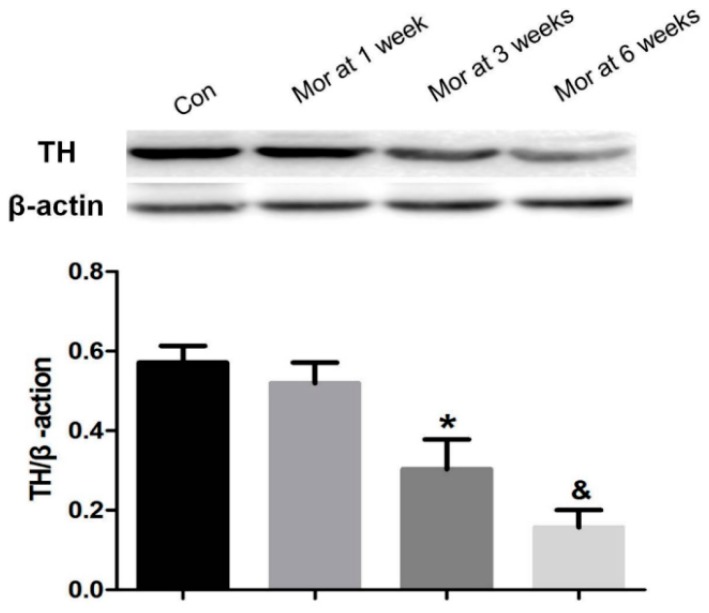
Western blotting analysis shows the expression of TH in the VTA. The expression of TH significantly decreased after three and six weeks of morphine dependence. Data are presented as the mean ± SEM (*n* = 4). * *P* < 0.05, **^&^**
*P*< 0.01 vs. control group.

**Figure 4 ijms-20-00250-f004:**
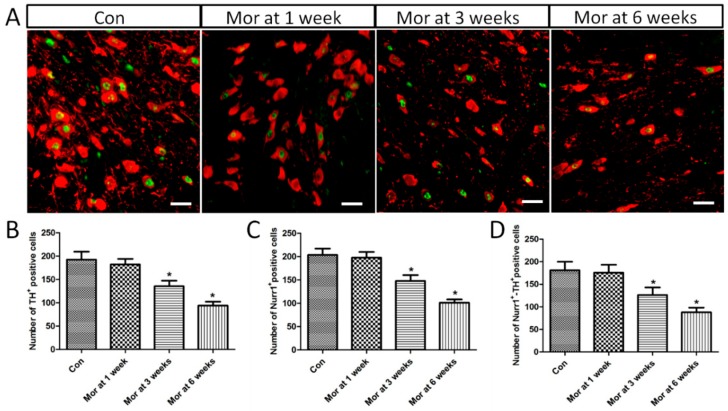
(**A**) Representative images showing the co-localization of Nurr1 and TH positive cells expression in the VTA. Bars = 25 μm. (**B**–**D**) Quantitative analysis of TH^+^, Nurr1^+^ and Nurr1^+^-TH^+^ positive cells. The data are shown as mean ± SEM, * *P* < 0.05 vs. control group (*n* = 8).

**Figure 5 ijms-20-00250-f005:**
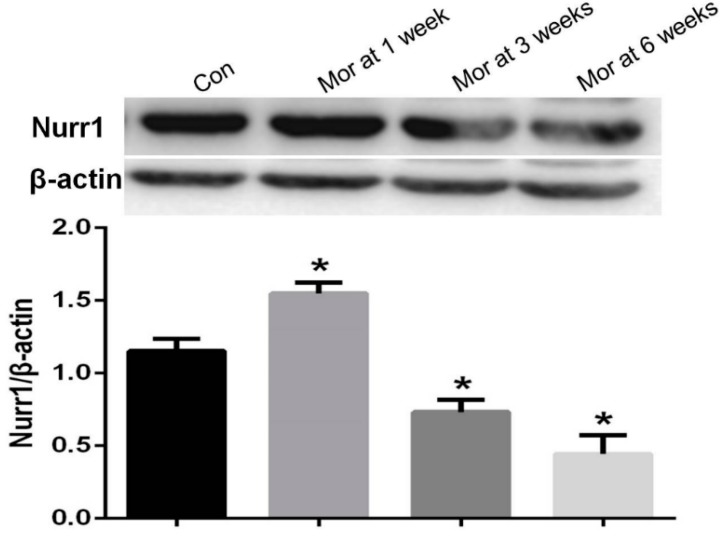
Western blot analysis of Nurr1 expression in the VTA. The relative level of Nurr1 in the VTA after one week of morphine exposure increased compared with the control group (*P* < 0.05). However, with prolonged morphine exposure, the relative Nurr1 expression of the three week and six week morphine-dependent groups decreased markedly (*P* < 0.05). Data are presented as the mean ± SEM. * *P* < 0.05 vs. control group (*n* = 4).

**Figure 6 ijms-20-00250-f006:**
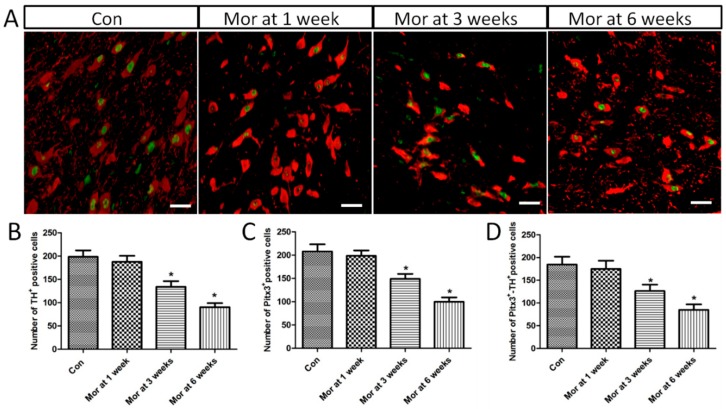
(**A**) Representative images showing the co-localization of Pitx3 and TH positive cell expression in the VTA. Bars = 25 μm. (**B**–**D**) Quantitative analysis of TH^+^, Pitx3^+^ and Pitx3^+^-TH^+^ positive cells. The data are shown as mean ± SEM. * *P* < 0.05 vs. control group (*n* = 8).

**Figure 7 ijms-20-00250-f007:**
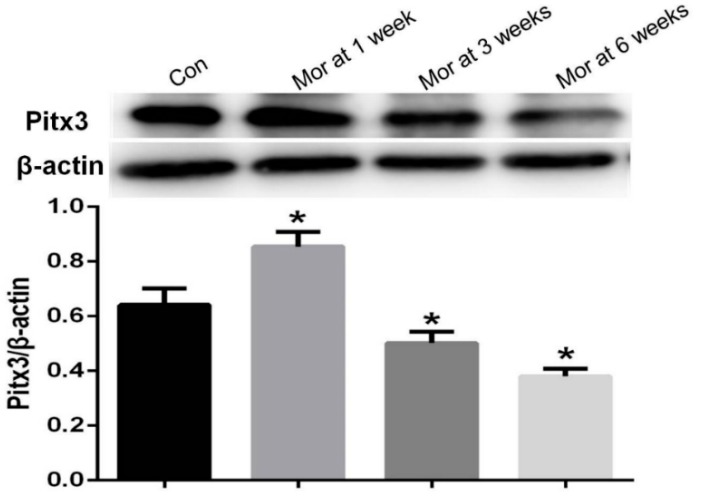
Western blot analysis of Pitx3 expression in the VTA. The relative level of Pitx3 was significantly up-regulated after one week of morphine dependence (*P* < 0.05). However, with prolonged morphine exposure, the level of Pitx3 was significantly decreased at three and six weeks (*P* < 0.05). Data are presented as the mean ± SEM. * *P* < 0.05 vs. control group (*n* = 4).

**Figure 8 ijms-20-00250-f008:**
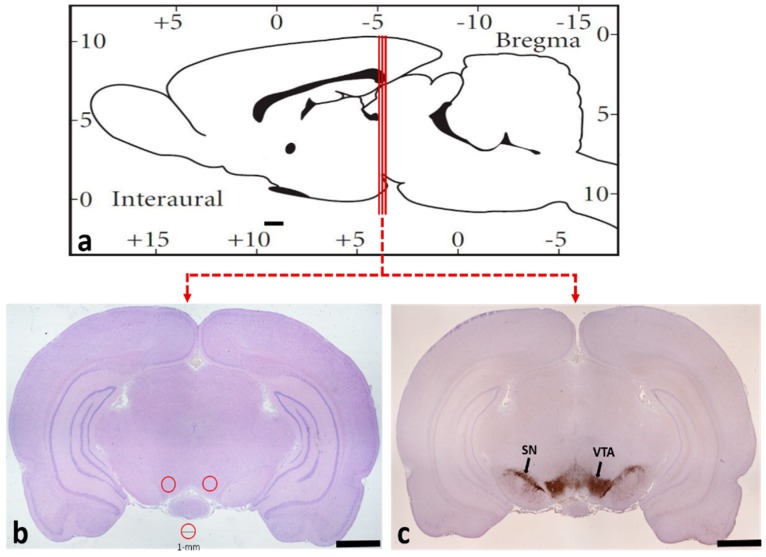
(**a**) The locations of the razor blades used to cut two adjacent VTA coronal slices are indicated in a sagittal view of the rat brain; (**b**) the punch placement in the consecutive coronal sections of the VTA (red circles); (**c**) the section with the largest area of the VTA. Bars = 1.6 mm in (**b**–**c**).

**Table 1 ijms-20-00250-t001:** Results of the count of withdrawal symptoms.

Symptom	Morphine-Dependent Group	Control Group
wet dog shakes	8.35 ± 1.86 *	0.74 ± 0.61
stretching	15.52 ± 4.23 **	0.63 ± 0.50
cleaning fur	4.96 ± 1.59 *	0.76 ± 0.53
swallowing	11.75 ± 1.78 **	0.89 ± 0.83
standing	8.00 ± 0.86 **	0.88 ± 0.63
jumping	3.45 ± 1.28 **	0.60 ± 0.76
teeth chattering	7.30 ± 1.78 **	0

* *P* < 0.05, ** *P* < 0.01 compared with the control group.

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
