# Peer review of "Dysregulation of Dopaminergic Regulatory Factors TH, Nurr1, and Pitx3 in the Ventral Tegmental Area Associated with Neuronal Injury Induced by Chronic Morphine Dependence"

_ijms, 2019, doi:10.3390/ijms20020250_

Reviewer 1 Report

The manuscript by Shi et al describes the effect of chronic morphine injection on VTA dopaminergic neurons in rat model. The authors describe a relationship between changes in the TH, Nurr1, and Pitx3n expression and neuronal injury.  Although the topic is interesting, it has already been reported by others (García-Pérez et al, Addict Biol. 2016 Mar;21(2):374-86). It is my opinion that the results reported by Shi et al.  don’t add relevant and new information in this field. For this reason, I believe that the manuscript may be not considered for publication

Author Response

Thanks for the reviewer’s comments. After careful reading the paper you mentioned (García-Pérez et al, Addict Biol. 2016 Mar;21(2):374-86), we found that some of the results were the same, which made our article less innovative. However, our research is different from that research to a certain extent, and still have certain novelty.

1. In order to simulate the characteristics of humans with long-term morphine dependence, we extended the time of chronic morphine dependent rat model to 6 weeks (only 1 week for García-Pérez’s paper). There was no significant changes in VTA dopamine neurons (TH) at 1 week, and this result was consistent with the study of García-Pérez. But after one week, the damage of dopamine neurons became more obvious with the prolongation of morphine exposure and the number of TH positive cells decreased, which are completely absent in García-Pérez’s paper. 

2. Nurr1 and Pitx3 are important regulators of dopamine neuron survival, which regulate the expression of TH. The results of our study showed that the expression of Nurr1 and Pitx3 decreased from 3 weeks of morphine dependence and could not effectively regulate the expression of TH. Thereby, the expression of TH was also significantly reduced. The decrease of these important factors of dopamine maintenance and survival seriously affected the survival of dopamine neurons, which may be one of the important reasons for the VTA dopamine neuron damage and the number of significant reduction. These findings are novel and not reflected in García-Pérez’s research.

In views of these above, sincerely hope to be able to give us opportunity to publish. Thank you so much.

Reviewer 2 Report

The research article entitled “Dysregulation of dopaminergic regulatory factors

TH, Nurr1, and Pitx3 in the ventral tegmental area is associated with neuronal injury induced by chronic morphine dependence” by Shi and co-workers investigate the function of several regulatory factors, such as TH, Nurr1, and Pitx3, on dopaminergic neurons viability and survival in a critical area involved in the development and maintenance of addiction, ventral tegmental area (VTA), after a long period of morphine dependence. The authors concluded that dysregulation of these regulatory factors in the VTA is associated with neuronal injury induced by chronic morphine dependence. The findings of the present study provide new insights on dopaminergic neuron damage.

The article is interesting, the design, methodology and results are clear.

Minor points:

 Page 2 line 46: delete point- “physiology. [9, 10, 11].” By “physiology [9, 10, 11].”

 Page 2 line 46: “play a critical role” instead of “play critical roles”

 Page 2 line 47: “is involved” instead of “plays a critical role”

 Page 3 line 109: “Experiments were performed according to manufacturer's instructions” instead of “In accordance with the protocol recommended by the immunohistochemistry kit”.

 Page 10 line 232: “relation” instead of “combination”.

 Page 10 line 244: “observed” instead of “seen”. 

 Page 11 line 271: “think” instead of “consider”. 

Author Response

Page 2 line 46: delete point- “physiology. [9, 10, 11].” By “physiology [9, 10, 11].”

     Page 2 line 46: “play a critical role” instead of “play critical roles”

     Page 2 line 47: “is involved” instead of “plays a critical role”

     Page 3 line 109: “Experiments were performed according to manufacturer's instructions” instead of “In accordance with the protocol recommended by the immunohistochemistry kit”.

     Page 10 line 232: “relation” instead of “combination”.

     Page 10 line 244: “observed” instead of “seen”. 

     Page 11 line 271: “think” instead of “consider”. 

 Thanks for the reviewer’s comments. These have been modified according to the reviewer’s opinion.

Round  2

Reviewer 1 Report

Although some results have already been reported (García-Pérez et al, Addict Biol. 2016 Mar;21(2):374-86), Shi et al describes the effect of longer chronic dependent  of morphine (6 week) instead 1 week reported by García-Pérez et al. The decreased expression of Nurr1 and Pitx3 after 3 weeks of morphine dependent and the reduced ability  to regulate TH expression is the novel finding reported in the manuscript by Shi et al. For this reason, I believe that the manuscript may be considered for publication with minor modification.

Page 2 line 46: “play a critical role” instead of “play critical roles”

 Page 2 line 47: “is involved” instead of “plays a critical role”

Page 10 line 244: “observed” instead of “seen”.

 Page 11 line 271: “think” instead of “consider”.

 Author Response

Page 2 line 46: “play a critical role” instead of “play critical roles”

Page 2 line 47: “is involved” instead of “plays a critical role”

Page 10 line 244: “observed” instead of “seen”.

Page 11 line 271: “think” instead of “consider”.

Thanks for the reviewer's comments. These have been modified according to the reviewer's opinion.
